# Human In Vitro Models of Epilepsy Using Embryonic and Induced Pluripotent Stem Cells

**DOI:** 10.3390/cells11243957

**Published:** 2022-12-07

**Authors:** Muhammad Shahid Javaid, Tracie Tan, Naomi Dvir, Alison Anderson, Terence J. O’Brien, Patrick Kwan, Ana Antonic-Baker

**Affiliations:** Department of Neuroscience, Central Clinical School, Monash University, Melbourne, VIC 3004, Australia

**Keywords:** stem cell lines, epilepsy, patient-specific cell models, neuronal differentiation, personalised disease models, personalised medicine

## Abstract

The challenges in making animal models of complex human epilepsy phenotypes with varied aetiology highlights the need to develop alternative disease models that can address the limitations of animal models by effectively recapitulating human pathophysiology. The advances in stem cell technology provide an opportunity to use human iPSCs to make disease-in-a-dish models. The focus of this review is to report the current information and progress in the generation of epileptic patient-specific iPSCs lines, isogenic control cell lines, and neuronal models. These in vitro models can be used to study the underlying pathological mechanisms of epilepsies, anti-seizure medication resistance, and can also be used for drug testing and drug screening with their isogenic control cell lines.

## 1. Introduction

Epilepsy affects 70 million people worldwide [1,2], causing significant morbidity and mortality. with more than half of the affected people living in countries with poor medical resources and little or no access to treatment [3]. It is characterised by unprovoked, recurrent seizures which result from the disruption in the balance between neuronal excitation and inhibition in the brain [1]. In most cases the cause of epilepsy is unknown but both genetic and environmental factors are understood to contribute to its aetiology. 

Genetic epilepsy is characterised by seizures that are the result of known genetic variance in one or many genes associated with epilepsy [2]. Mutations in several genes encoding ion channels and proteins have been reported to be most commonly associated with epilepsy. These genes include, but are not limited to mutation in *SCN1A* (encode sodium channel protein), which is associated with Dravat syndrome; *KCNQ2* or *KCNQ3* (both encoding potassium channel protein), which are associated with benign neuronal familial seizures [3,4,5]; the *CHRNA4* gene (20q13), which is associated with Autosomal Dominant Nocturnal Frontal Lobe Epilepsy (ADNFLE), characterised by hypermotor nocturnal seizures [6]; and the mammalian target of rapamycin (mTOR) pathway, in particular *DEPDC5* gene of this pathway, which are associated with Focal Cortical Dysplasia (FCD) type IIa and IIb [7,8,9,10,11,12,13,14,15].

Anti-seizure medications (ASMs) are the mainstay of treatments for epilepsy. Despite multiple newly developed drug treatments being introduced to clinical practice, more than 30% of epileptic patients remain drug-resistant [16]. Genetic causes account for almost 20% of drug-resistant epilepsy cases in children. Surgery may be the only curative treatment option for these refractory epilepsy patients. However, ASMs and surgery are not always successful due to an incomplete understanding of epilepsy aetiology and pathogenesis resulting in non-targeted treatments [16,17,18,19]. There is, therefore, a need for in-depth investigations to gain a better understanding of the pathological mechanisms, an understanding which would inform the development of new treatments.

Currently, potential ASMs are validated using acute-seizure animal models prior to clinical development. However, animal models of genetic epilepsy have several limitations [20,21,22,23]. The major concern is the species-specific differences [23] leading to differences in physiological development and lack of human-specific receptors and drug targets. Hence, there is a demanding need for human-based disease models to develop new therapeutic strategies to achieve seizure freedom for these patients.

The development of human-based in vitro disease models [24] is an active area of research. One potential source of human in vitro models is the use of human pluripotent stem cells. These models can either be derived from human embryonic stem cells (hESCs) or induced pluripotent stem cells (iPSCs). The iPSCs derived from healthy individuals or patients are a promising approach to developing regenerative therapies as well as in vitro models of pathophysiological mechanisms of diseases (Figure 1) [25]. 

These models better recapitulate the complexity of genetic epilepsy. In recent years, stem cell research has focused on using patient-specific hiPSC-derived neurons as in vitro models of epilepsy.

This scoping review aims to assess the application of stem cell-derived in vitro models to model the pathophysiology of epilepsy in the current literature. Specifically, we will determine: (i) what different methodologies have been used to generate in vitro epilepsy models; (ii) which gene variants and specific epileptic phenotypes have been modelled, and (iii) which outcome measures have been investigated.

## 2. Methods

This review was conducted in accordance with the Preferred Reporting Items for Scoping Review (PRISMA-ScR) guidelines (Figure 2) [26]. To minimise the selection bias that is often associated with narrative reviews, we employed the same rigorous methodology used in performing a systematic review.

### 2.1. Search Strategies

Three electronic databases were searched (Web of Science, PubMed, EMBASE). The predetermined searched terms were as follows:

**Search terms for epilepsy:”** Epilepsy” or “Refractory epilepsy” or “Seizures” or “Idiopathic generalized epilepsy” or “IGE” or “Frontal lobe epilepsy” or “FLE” or “Temporal lobe epilepsy” or “TLE” or “Autosomal dominant nocturnal frontal temporal lobe epilepsy” or “ADNFLE” or “Familial temporal lobe epilepsy” or “FTLE” or “Genetic Epilepsy” or “Chemically induced epilepsy”.

**Search terms for stem cells:** “Induced Pluripotent Stem Cells” or “iPSCs” or “Pluripotent Stem Cells” or “Human induced pluripotent stem cells” or “hiPSCs” or “Embryonic stem cells” or “Embryonic derived stem cells” or “Human embryonic stem cells” or “ESCs”. Appropriate search symbols and Boolean operators were used to combine both lists. 

### 2.2. Eligibility Criteria

In this scoping review, all original studies that used stem cells or stem cell-derived cells as disease models to study epilepsy were included. The literature search was performed in 2018 and then updated in 2021. No year limit was set to include all the published studies as the focus was to assess the progress of the stem cell field in epilepsy research. The studies were not limited to a particular outcome measure or the method of neuronal induction. We excluded studies that did not use stem cells, did not assess epilepsy, and did not contain original data, such as reviews, letters, and commentaries.

### 2.3. Publication Selection

All publications were screened against our predetermined inclusion and exclusion criteria. As per scoping review guidelines, all retrieved publications were screened first by title and abstract followed by a full-text screen, by at least two independent reviewers (MSJ and TT). Any discrepancy between reviewers was resolved by a third reviewer (AAB).

Information was extracted from the selected studies using a standard data extraction template including the epilepsy phenotypes/models of epilepsy based on genetic variants (duplication or deletion of chromosomes), starting cell source/cells type, stem cell types, the cell population used in the study, gene mutation, and the main outcomes of the included studies (Appendix A).

### 2.4. Information Extraction

Data were extracted from the papers included in the scoping review by two independent reviewers (M.S.J. and T.T.) using a data extraction tool developed by the reviewers. A draft extraction form is provided (Appendix A). Any disagreements that arose between the reviewers were resolved through discussion, and with an additional reviewer (A.A.B.). The extracted data included specific details about the participants, concept, context, study methods, and key findings relevant to the review question.

### 2.5. Data Analysis and Presentation

The findings are presented graphically (Figure 3 and Figure 4), diagrammatically (Figure 1), and in tabular form (Appendix A). The tabulated and/or charted results were discussed to provide a narrative summary of the review’s question and objective. 

## 3. Results

### 3.1. Study Selection Process

The major findings of this study are summarized in Appendix A. A total of 2997 studies were identified, of which 1106 were removed as duplicates. The remaining 1889 were screened by title and abstract, and 1677 were excluded. The remaining 212 full-text studies were then reviewed, and 123 were excluded based on the inclusion/exclusion criteria. Full-text screening yielded 89 studies that met our pre-specified inclusion criteria (Figure 2). Of the included studies, human embryonic stem cells (hESCs) were used in 16 studies, and human induced pluripotent stem cells (hiPSCs) were used in the remainder (Figure 3). 

The studies using hESCs-derived neurons investigated: their differentiation potential to generate specific neuronal subtypes; how genetic variance affects neuronal behaviour and how it influences the development of neuronal diseases such as FOXG1 syndrome and TSC [27,28,29,30,31,32,33,34,35,36,37]. In particular, a study performed by Costa et al, assessed how neurodevelopment and synaptic plasticity were altered in response to mTORC1 inhibition in tuberous sclerosis (TSC) [35]. In other studies, hESCs-derived neurons have been used: to model FOXG1 syndrome to control the endogenous protein dosage of FOXG1 protein in a precise manner which is important for GABA interneuron differentiation [37]; to assess the effect of high glucose concentration in masking the TSC cellular phenotypes using a hESCs-derived TSC model [38]; to investigate genotoxicity of anti-seizure medications [39], and the mode of action of the ASM valproate [40]. Two of these studies also used genetically engineered cells to enhance the release and delivery of adenosine [41,42]. 

In this review, the studies using hiPSCs were grouped into two major categories. The first group is the studies that described the generation of patient-specific iPSC lines [43,44,45,46,47,48,49,50,51,52,53,54,55,56] that carry genetic variants in genes associated with epilepsy, such as *SCN1A*, *GNB5, LGI1, GRIN2A, KCNC1,* or *KCNA2* (Figure 3, Appendix A) [43,44,45,46,47,48,49,50,51,52,53,54,55,56]. The second group is the studies that assessed the effects of convulsant and anticonvulsant drugs on hiPSC-derived neurons and astrocytes (Figure 4, Appendix A) [57,58,59]. Some of these studies used healthy human stem cells to either generate an in vitro disease model by genetically manipulating a gene of interest; or by assessing the effects of different compounds on chemically induced “epilepsy-like” phenotype on otherwise healthy neurons [57,58,59,60]. Others used the iPSCs derived from patients carrying a specific gene variance of interest (Figure 4) [27,36,38,43,44,45,46,47,48,49,50,51,52,53,54,55,56,57,58,60,61,62,63,64,65,66,67,68,69,70,71,72,73,74,75,76,77,78,79,80,81,82,83,84,85,86,87,88,89,90,91,92,93,94,95,96,97,98,99,100,101,102,103,104,105,106,107,108,109,110,111,112].

### 3.2. Gene Editing Techniques for Disease Modelling and Isogenic Control

The methodology used to induce genetic variance in healthy cells for disease modelling, or to correct patient cells for isogenic control, has evolved over time. Sixteen studies (Appendix A) reported the use of gene editing techniques to create isogenic controls or to study the disease-causing genetic variation in patient-derived iPSCs. Transcription Activator-Like Effector Nucleases (TALEN) was one of the first gene-editing techniques used in 2014 to generate the *SCN1A* mutation in human iPSCs [61] and was subsequently used in two more studies [48,55]. Zinc-Finger Nucleases (ZFNs) is another gene-editing technology and was used to model TSC in 2016 [35]. In addition, virus- or vector-based knock-out and knock-down techniques were also used in a few studies [33,42,95,106]. These methods were subsequently replaced with the more advanced CRISPR/Cas9 method. Unlike its predecessors, TALEN and ZFNs, it is precise, robust, and site-specific with fewer off-target effects [113]. The first use of CRISPR/Cas9 technology in epilepsy was reported in 2016 to generate a loss of function *SCN1A* mutation in human iPSCs to gain insight into Dravet syndrome [114]. This approach has been widely adopted in recent years with eight additional studies generating disease-specific neurons and isogenic controls using CRISPR/Cas9 [37,44,68,70,74,97,112,114].

### 3.3. Epilepsy Patient-Specific iPSCs Derived Disease Models

Dravet syndrome was the most commonly studied [60,61,62,63,64,65,67,68,69,114], followed by tuberous sclerosis [35,38,70,71,72], focal cortical dysplasia [82,83,84,85,104], and a few rare epilepsy syndromes [44,45,46,47,53,73,74,76,77,79,80,81,87,115]. These models have the potential to provide useful information about the involvement of a particular gene in disease progression and its anticonvulsant response.

The first in vitro model from a Dravet patient carrying a mutation in the *SCN1A* gene was generated in 2013. The findings from that study showed that the loss of function in GABAergic inhibition appears to be the main driver in epileptogenesis [62]. Since then, there have been several studies assessing various mutations in the *SCN1A* gene [48,54,55,56,60,61,62,63,64,65,66,67,68,114]. Neurons derived from these patients exhibit increased sodium currents and hyperexcitability [64], which can be alleviated by treatment with phenytoin [60]. In another study, neurons derived from two patients with Dravet syndrome demonstrated that genetic alterations of *SCN1A* differentially impacted electrophysiological impairment. The degree of impairment corresponded with the symptomatic severity of the donor from which the iPSCs were derived [63]. Recently, another patient-specific iPSCs-derived neuronal study generated from individuals with *SCN1A* mutation indicated an imbalance in excitation and inhibition that leads to hyperactivity in the neural network. This study used homozygous and isogenic controls to show the hyperexcitability in the generated neurons [68]. These studies indicated that neurons could recapitulate the neuronal pathophysiology and could potentially be used for screening drugs for personalised therapies [60,64]. 

The most commonly studied mTORopathies were tuberous sclerosis and focal cortical dysplasia (FCD). A study using neurons derived from a patient with *TSC2* mutation reported hyperactivation of mTORC1 pathway [35]. In this model, pharmacological inhibition of mTORC1 with rapamycin reverses developmental abnormalities and synaptic dysfunction during independent developmental stages [35]. In another study, neuronal progenitor cells (NPCs) generated from a patient carrying a heterozygous *TSC2* mutation exhibited disrupted neuronal development, potentially contributing to the disease neuropathology. Moreover, NPCs also exhibited activation of mTORC1 downstream signalling and attenuation of PI3K/AKT signalling upstream of TSC [72]. More recently, NPCs generated from the patient carrying TSC germline nonsense mutation in exon 15 of *TSC1* showed the influence of *TSC1* mutation in the early neurodevelopmental phenotypes, signalling, and gene expression in NPCs compared to the genetically matched wild-type cells [70]. In 2019, Sundberg et al showed that loss of one allele of *TSC2* is sufficient to cause some morphological and physiological changes, elevated phosphorylation, and hyperexcitability of mTORC1 in human neurons, but biallelic mutations in *TSC2* are necessary to induce gene expression dysregulation seen in cortical tubers. They also found that treatment of *TSC2* patient-specific iPSCs-derived neurons with rapamycin reduced neuronal activity and partially reversed gene expression abnormalities [71]. In 2020, Alsaqati used commercially available *TSC2* (loss of function mutation) patient-derived iPSCs and reported that the dysfunctional neuronal network behaviour in the differentiated neurons could not be rescued with rapamycin treatment [57]. The difference in response in these two studies is because of two different iPSCs samples carrying different mutations in the *TSC2* gene [57,71]. 

Five studies assessed the FCD-related cortical malformation by generating iPSCs from patients with mutations in genes involved in regulating the mTOR pathway. In 2017, Marinowic et al described the generation of iPSC-based cellular models of refractory epilepsy from the fibroblasts of two refractory epilepsy patients with FCD type IIb, one a 45-year-old male and the other 12-years-old female [84]. Then in 2020, Marinowic published another study using these cells investigating the differences in the migration potential and the expression of genes for cell proliferation, adhesion, and apoptosis. The main finding of the study was that the gene expression was different between the neurons generated from the adult male compared to the child. They concluded that differences in the migration potential of adult cells, and differences in the expression of genes related to the fundamental brain development processes, might be associated with cortical alteration in the two patients with FCD IIb [85]. In 2018, Majolo et al studied the Notch signalling pathway, a pathway involved in cortical development to regulate neuronal differentiation, self-renewal, survival, and neuronal plasticity, using the iPSCs from FCD IIb patients. The study assess the expression of genes involved in Notch signalling and showed that, during embryonic neurogenesis, the neural precursor cells of FCD type IIb individuals exhibited an increase in *HEY1* and *NOTCH1* genes as well as a decrease in the expression of *HES1* and *PAX5* genes, compared to the cells from control subjects [82]. In the subsequent study, Majolo et al studied the migration and synaptic aspects of neurons generated using the iPSCs derived from patients with FCD type IIb. Using real-time PCR, the study presented the expression of most of the synaptic and ion channels genes *ASCL1, DCX, DLG4, FGF2, NEFL, NEUROD2, NEUROD6, NRCAM,* and *STX1A* in different groups; fibroblasts, iPSCs, differentiated neurons, and brain tissues [83]. This study suggested that the cells derived from FCD patients may have more sensitivity to stimuli resulting in altered cell survival, apoptosis, migration, and morphological development. In 2021, Klofas et al published a study using the FCD patient-derived neurons carrying a heterozygous loss-of-function mutation in the *DEPDC5* gene and reported hyperactivation of mTORC1 and enlarged cell somas that were rescued with the inhibition of mTORC1. This study also reported that cell starvation leads to hyperactivation of the mTOR pathway [104] but the exact mechanism is still unclear. None of these FCD studies have performed electrophysiological functional analysis of the generated neurons. 

### 3.4. Outcome Measures

Eighty-eight studies (98%) reported histological, molecular, electrophysiological, or other outcomes to validate the generation of iPSCs or to assess epilepsy-like phenotypes. 87% of studies (Appendix A) reported the molecular outcomes to validate the successful generation of patients’ iPSCs and iPSC-derived neuronal disease models, of which 38% (Appendix A) also reported the electrical activity of generated neurons using either a patch-clamp or Micro-Electrode Array (MEA) analysis to assess electrophysiological activity.

Electrophysiological analysis was performed using patch-clamp [27,32,33,35,37,60,61,62,63,64,66,67,68,69,77,78,89,90,94,96,97,98,103,107,110,114,116] or MEA [34,36,57,59,69,71,75,86,93,105,108] analysis. Electrophysiology is a preferred method of studying brain activity because it allows the recording of a wide range of neuronal phenomena ranging from the action potential to the network simulation of a neuronal population [27,110,117,118,119]. Patch-clamp or MEA can also be used to measure intracellular voltage [120].

Real-time PCR, western blot, immunoblot, and immunofluorescence were the primary molecular and histological methods applied to validate patient iPSC-derived neuronal disease models. However, in the case of patient-specific iPSC generation carrying a specific genetic variant, Sanger sequencing was the primary measure to confirm the presence of the variant (Appendix A). For cell lines, these outcome measures are a standard requirement to publish and report the generation of cell lines. Similarly, for disease modelling research, standardised outcome measures for functional and phenotypical analysis of the generated neurons are needed.

## 4. Discussion

Identifying and understanding the pathological mechanisms of human diseases such as epilepsy play a crucial role in the development of novel therapeutic approaches. Unfortunately, currently available anti-seizure medications are unable to treat epilepsy in about 30% of patients. The limited efficacy of these ASMs has been, at least partly, attributed to the lack of appropriate pre-clinical models. Most of the in vivo and in vitro disease modelling and drug development is done in animals, and however useful, these models are not ideal to study genetic epilepsy [23]. Some attempts have been made to utilise primary brain tissue from patients, but its limited availability and the difficulty of culturing it makes it challenging to use. Therefore, the ability to utilise patient cells and generate iPSC-derived neurons is of great value to the field of neuroscience. In this review, we have summarised the current literature describing stem cell-derived in vitro epilepsy models. 

Using human stem cells to model epilepsy in vitro is a relatively new concept, with the first study published in 2004. Since then, however, the field has exponentially grown with over 25 studies published in the past three years (Appendix A). While the early publications reported artificially induced epilepsy-like phenotypes in hESCs, the more recent publications described patient-specific iPSC lines that carry a disease-causing gene variant.

We identified 65 studies describing the establishment of new patient-derived iPSCs lines. While some studies only reported the generation and characterization of these lines, the majority (50 studies) used hPSCs to assess the involvement of a particular gene in disease progression and drug response. These studies have shown that neurons derived from the patient iPSCs are phenotypically and morphologically different compared to healthy neurons or neurons derived from isogenic controls. Moreover, these neurons exhibited delayed differentiation, synaptic abnormalities, and defects in neurite formation and migration [87,95]. They also have a unique electrophysiological signature that differs between both, the individual patients and the controls. Transcriptional changes and disrupted pathways of chromatic modelling specific to a gene variant have also been revealed using patient-specific iPSCs-derived neurons [66,78]. Furthermore, these neurons exhibit bursting impairments leading to hyperpolarization and hyperexcitability which can be altered by the administration of different ASMs. In vitro models generated from patient-derived neurons provide new insight into the disease phenotype, and molecular and cellular mechanisms that underlie epileptogenesis and drug resistance in individual patients, thus identifying crucial pathways of drug screening for the development of novel anti-seizure medications, and precision, and regenerative medicines.

### 4.1. Limitations of Stem Cell-Derived Epilepsy Models

The iPSCs models hold potential opportunities to enhance our current understanding across a wide range of biological phenomena in epilepsy and beyond, but they come with limitations.

No two iPSC lines or models derived from these lines are the same. They are as unique as the individual from whom they are derived. In addition, protocols used to generate these cells also vary. There is a degree of experimental variability associated with the generation of iPSCs [121], including the source of somatic cells from which the iPSCs were generated, initial cell density used to generate iPSCs, and variability in the reprogramming kits as well as time and duration of each step. To combat these issues, the International Society for Stem Cell Research (ISSCR) has developed a set of guidelines for the generation of new iPSCs lines (https://www.isscr.org/policy/guidelines-for-stem-cell-research-and-clinical-translation/sections/part5. Assessed on 8 August 2022). These guidelines require each newly generated iPSC line to be registered and characterised using pluripotency assays, teratoma formation analysis, germ layer differentiation, donor screening, variant analysis, sequencing, karyotyping, and mycoplasma detection analysis (https://www.journals.elsevier.com/stem-cell-research/lab-resources/scientific-guidelines-for-lab-resources. Assessed on 8 August 2022).

Another source of variability is the protocols used to generate neuronal cultures from these iPSCs. Despite there being only two main methods of neuronal differentiation (either dual SMAD inhibition by using small molecules [122] or the viral vectors method [123]), there are a number of different agents used and each different protocol yields different proportions of neuronal and glial cells. The most physiologically relevant method uses the dual SMAD inhibition which gives rise to both excitatory and inhibitory neuronal populations as well as a small proportion of astrocytes, which are all required for the generation of functional and neuronal networks in vitro. However, this method is time-consuming and costly as it takes several months to generate mature neurons. The alternative is to use viral vector transduction into stem cells to express specific genes for neuronal differentiation. In this method, stem cells are modified to overexpress only one or two neuronal genes, and therefore produce more homogeneous cultures and, in many cases, do not contain astrocytes. As the astrocytes are required for the formation and maintenance of mature functional neuronal networks [124], these methods require co-culturing with an external source of astrocytes. These astrocytes are, however, generated from healthy human brains and do not recapitulate the epileptic phenotype. For the generation of in vitro models that can effectively model epileptogenesis, it is critical to generate both patient-derived excitatory and inhibitory neuronal populations as well as glia. 

For the better characterisation and reproducibility of in vitro models, either for the study of the pathophysiology of epilepsy or for drug screening, the methodology of neuronal differentiation and functional assessment need to be standardised. In addition, the sharing of data amongst researchers would allow for the comparison and standardisation of protocols which would maximise reproducibility across different labs. Highly curated and referenced cell lines should be available to the scientific community in the form of cell banks. A few cell banks are: Stem Cells for Biological Assays of Novel Drugs and Predictive Toxicology (StemBANCC) [125], HipSci [126], the European Bank for Induced Pluripotent Stem Cells (EBiSC) [127], and the iPSC Collection for Omic Research (iPSCORE) [128]. A similar standardised depository of neuronal differentiation methods and neuronal functional outcome measures is required to enhance reproducibility and increase the likelihood of developing new and novel therapies.

### 4.2. Limitations of Our Approach

The major limitation of this study was the sheer magnitude of variability of the studies included. Due to the broad nature of our research questions, the findings of this review are similarly broad. The idea of using stem cell-derived neurons as human in vitro models of epilepsy is a relatively new concept and therefore we did not want to limit our search to a particular treatment or outcome measure. This meant that our included dataset contained a variety of different studies with different study designs and outcome measures.

Due to the lack of standardised reporting of human in vitro models, it was difficult to perform any type of risk of bias assessment. Studies varied in the methods of iPSC generation, neuronal differentiation, and outcome measures. Only thirteen studies assessed the effects of potential novel therapies (Appendix A). There have been some efforts made in developing risk bias tools for in vitro studies [129], but for those tools to be applicable for the assessment of in vitro epilepsy models, the methodology and reporting of such models would need to be standardised.

## 5. Conclusions

The stem cell field has tremendous potential to revolutionise both preclinical and clinical epilepsy research. Advancements in iPSC reprogramming, differentiation, and genome engineering have expanded the use of cell-based models into the mainstream of cellular neuroscience. As the field of in vitro disease modelling evolved, it is critical to develop and standardise methodologies to improve translatability, integrity, and quality [130]. Rigorous optimisation to standardise the approaches may take extra effort, but is an essential requirement to translate the promise of stem cell-based disease models to the development of new therapeutics and therapies. Personalised disease models would allow the neurobiologist to investigate the unknown details of the disease processes by mimicking human pathophysiology, which could lead to major advances in personalised medicines, and would help to address the issue of drug resistance.

## Figures and Tables

**Figure 1 cells-11-03957-f001:**
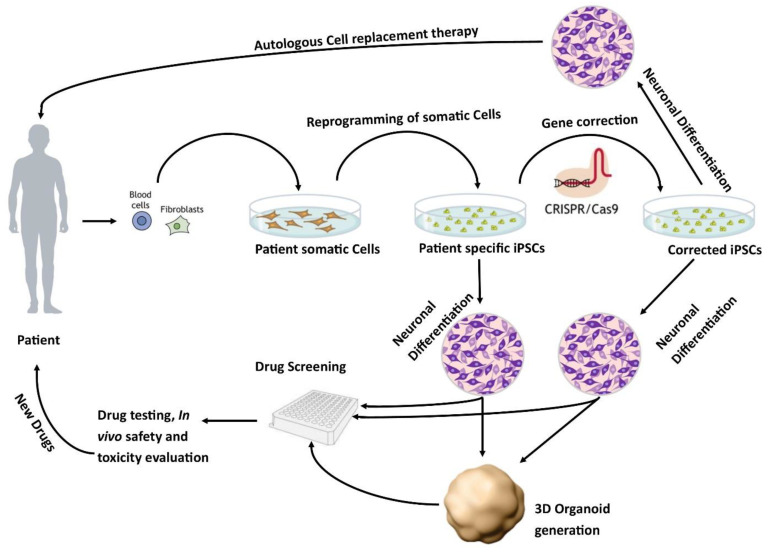
Patient-specific stem cell lines for cell replacement therapy and new drug development. Somatic cells from the patient’s skin or blood can be isolated and reprogrammed into iPSCs. These cells have the potential to differentiate into different neuronal populations and can be used to either further study the pathophysiology of the disease or as a drug screening system to test new potential therapies. In addition, these lines can be corrected using CRISPR/Cas9 to generate isogenic controls which provide ideal controls.

**Figure 2 cells-11-03957-f002:**
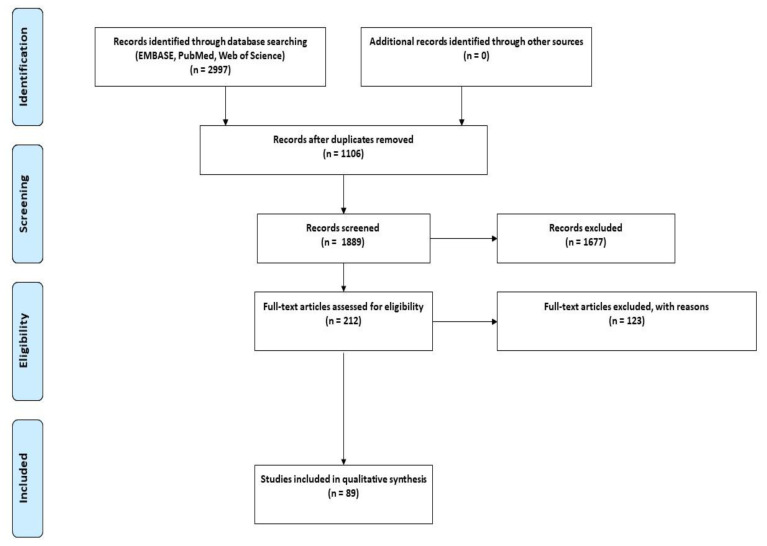
PRISMA flow diagram of databases search, two-phase screening, and data extraction workflow. From 2997 studies, 1889 studies were screened for the title and abstracts. Of these, 212 studies underwent a full-text screen which identified the 89 studies that were included in the review.

**Figure 3 cells-11-03957-f003:**
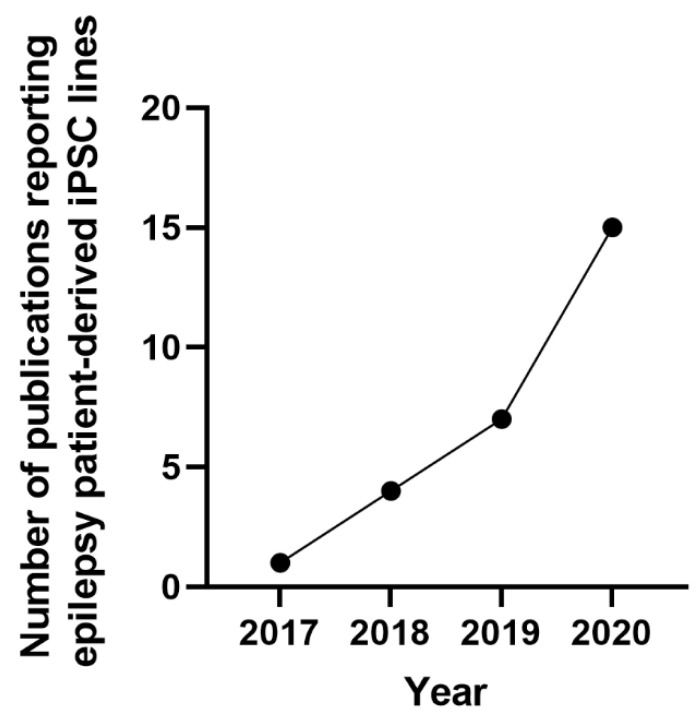
Generation of epileptic patient-specific iPSC lines. The graph indicates the increase in the number of publications reported from 2017 to 2020 that generated iPSC lines from patients with epilepsy.

**Figure 4 cells-11-03957-f004:**
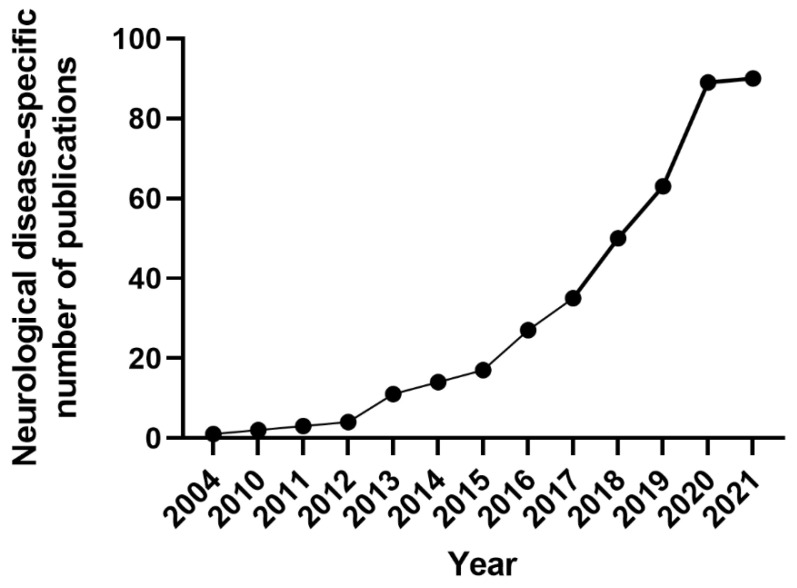
Publications since 2004 in the stem cell field for in vitro epilepsy modelling and drug toxicity testing using patient-derived iPSCs and human ESCs. The graph represents the total number of original research articles published in the field of epilepsy from 2004 to 2021 using stem cells.

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
