# Peer review of "Human In Vitro Models of Epilepsy Using Embryonic and Induced Pluripotent Stem Cells"

_cells, 2022, doi:10.3390/cells11243957_

Round 1
Reviewer 1 Report
This review by Javaid and colleagues does a very good job at their scoping review of studies using stem cells as disease model to study epilepsy. They collected references related to different methodologies to generate the stem cell models, they cataloged specific gene variants and epileptic phenotypes, and assess outcome measures of the tabulated studies. They provide a sufficient introduction to the field of stem cells and the generation of induced pluripotent stem cells (iPSCs) related to epilepsy. The authors give a comprehensive collection of cataloged references (Table1). The extensive collection of references and the compilation of studies completed to date indicating epilepsy phenotype, stem cell type, cellular manipulation, gene variant, and outcome measures will be a useful resource and important addition to the literature within the field.
From these tabulated references, the authors compare the use of human embryonic stem cells (hESCs) and iPSCs (Figure 3), with highlights from particular studies related to specific syndromes. They then compare patient specific iPSCs with known genetic variants associated with epilepsy; effect of different drugs on iPSCs derived neurons and astrocytes, modified genes from health cells, and iPSCs from patients with different gene variants.
They go on to make recommendations for the field for standardizing outcome measures for patient-specific cell models to show epilepsy phenotypes and were focused especially on electrophysiology analysis via patch-clam or micro-electrode array. The authors explain the standardization for the generation of iPSCs and conclude there must also be a standardization within the field of patient derived iPSCs to model epilepsy to account for variations in neuronal differentiation.
The authors also provide a robust discussion of patient derived cell lines throughout the review. They specifically discuss cell lines derived from patients with Dravet syndrome and tuberous sclerosis as the most common syndromes with patient derived cell lines. The authors review specific studies and findings, including involvements of particular genes and also importantly control cell lines. This is important as these cell lines will be important tools for “better understanding of the pathological mechanisms to inform the development of new treatments”.
I have a few specific comments regarding the manuscript and figures:
- The text in Figure 1 is slightly blurry.
- Figure 4 is referred to as Figure 3 in the figure legend.
- It does not appear there was a year limit to the authors initial search, although no searches were found before 2004 and the search was conducted in 2021. So, it would be nice to include this in the text as it is included in the Figure 4 legend. Lines 81-82 are where the authors discuss the eligibility criteria and studies included, and the date range would fit here nicely.
- In general, the references are relevant and current; however, the reference indicated in lines 142-147 of the manuscript do not match with all the references in Table 1. For example, on line 142 references [46-48] are indicated as part of Table 1, but reference 46 is not included in Table 1. Similarly, in line 147 references [15, 24, 26, 31-47, 50-103] are indicated, but references 44, 46, 58, 59, 61, 63, 98 and 102 are not included in Table 1. It is my recommendation that the authors double check the references indicated in these lines of the manuscript.
Author Response
We would like to thank the reviewer for their comments. By carefully considering the reviewer recommendations to our earlier submission, we decided to take into account those suggestions, and have now essentially produced a significantly enhanced manuscript. We have responded to the reviewer’s comments in a point-by-point manner and have uploaded our responses with the revised manuscript.

Reviewer 2 Report
This review aims at providing an overview of the in vitro models of epilepsy based on pluripotent stem cells. This is an interesting and relevant study with a comprehensive coverage of the literature on this field over the last years. Nevertheless, there are some major problems that hinder its relevance for publication in its present form:
1) The introduction lacks a contextualization about epilepsy pathophysiology and a description about different epilepsy phenotypes
2) The table provided is too extensive (25 pages), not well-organized and contains information that is not relevant for capturing the general picture of in vitro research models of epilepsy. Some suggested alterations are as follows:
a. Exclude Sr. No. , keep only ref number
b. Group by epilepsy phenotype
c. Include only human models (as indicated in the review title), thereby excluding this column, as well as the two studies with rodent cells
d. Cell manipulation is a confusing topic and the information contained in this column is not actually relevant as it is; this should be focused on describing the types of cells (types of neuron and/or glia) generated from iPSCs / hESCs
e. The indicated histological, molecular, and electrophysiological outcomes are not actually outcomes, but methodologies used for the characterization of iPSCs or in vitro phenotypes assessment. This is one of the most important topics to be correctly mentioned - the alterations identified in these in vitro models, how was gene, protein expression altered? Which genes/proteins? Which alterations were found at the electrophysiological level?
3) The information provided in discussion section is not appropriate for a discussion. It provides a summary of study goals and outcomes (that are once again more related to methodologies), which would be suitable for being presented in the introduction section. The discussion should be focused on exploring the different types of neurons and glia generated in these studies and how could they model the alterations identified in epilepsy patients.
4) The presented future challenges are in turn, more suitable for the discussion section. Moreover, it describes general problems of the field (e.g. iPSCs characterization and sample availability) rather than focusing on real challenges of in vitro research models in epilepsy field (e.g. have these studies generated models based on the most relevant cell types for epilepsy, is there a dimension of the disease that is not being addressed in these models?)
Author Response

(The authors gave the same response as above.)

Reviewer 3 Report
Good and comprehensive literature review on the in vitro cell models used for epilepsy research.
Some comments:
-Figure 1 needs a better resolution.
- Table 1 could go to supplementary data.
- The title of the y-axis in Figure 3 should be revised, there is a typo in Epilepsy and it should state precisely what is being measured.
- Showing the number of publications per year and the number of cell lines generated is not very relevant to the field. It would be more interesting to have figures that visually summarize the data in Table 1, for example.
Author Response

(The authors gave the same response as above.)

Round 2
Reviewer 2 Report
There have been major improvements in the introduction and in the table. Nevertheless, there are still some required revisions to unveil the potential of in vitro models for epilepsy research.
First, the discussion lacks focus on epilepsy in vitro models. As mentioned in previous revision, the topics discussed are too general, covering challenges of all pluripotent-derived neuronal in vitro models. The most important topics are presented in section 3.3 “Epilepsy patient-specific iPSCs derived disease models”. This should be the focus of the paper, and especially the discussion – to describe how have iPSC-derived in vitro models contributed for the understanding of epilepsy and what is still required in future studies.
Second, the column “outcome measures” is not a relevant topic for the table. It simply describes the techniques used, which can be simply due to the available techniques in the lab. This should target the outcomes of the studies, i.e. which cellular alterations were described in each study.
Author Response
We thank the reviewer for their valuable feedback. As suggested by the reviewer we edited Table S1 and included the main outcomes from each paper. Also, we have restructured the discussion section to better describe the main findings of our review.